# YTH-RNA-binding protein prevents deleterious expression of meiotic proteins by tethering their mRNAs to nuclear foci

Yuichi Shichino[1†], Yoko Otsubo[1], Yoshitaka Kimori[2,3,4], Masayuki Yamamoto[1,4], Akira Yamashita[1,4]*

[1]Laboratory of Cell Responses, National Institute for Basic Biology, Okazaki, Japan; [2]Department of Imaging Science, Center for Novel Science Initiatives, National Institutes of Natural Sciences, Okazaki, Japan; [3]Laboratory of Biological Diversity, National Institute for Basic Biology, Okazaki, Japan; [4]Department of Basic Biology, School of Life Science, SOKENDAI (The Graduate University for Advanced Studies), Okazaki, Japan

**Abstract** Accurate and extensive regulation of meiotic gene expression is crucial to distinguish germ cells from somatic cells. In the fission yeast *Schizosaccharomyces pombe,* a YTH family RNA-binding protein, Mmi1, directs the nuclear exosome-mediated elimination of meiotic transcripts during vegetative proliferation. Mmi1 also induces the formation of facultative heterochromatin at a subset of its target genes. Here, we show that Mmi1 prevents the mistimed expression of meiotic proteins by tethering their mRNAs to the nuclear foci. Mmi1 interacts with itself with the assistance of a homolog of Enhancer of Rudimentary, Erh1. Mmi1 self-interaction is required for foci formation, target transcript elimination, their nuclear retention, and protein expression inhibition. We propose that nuclear foci formed by Mmi1 are not only the site of RNA degradation, but also of sequestration of meiotic transcripts from the translation machinery.

DOI: https://doi.org/10.7554/eLife.32155.001

*For correspondence:
ymst@nibb.ac.jp

Present address: †RNA Systems Biochemistry Laboratory, RIKEN, Saitama, Japan

Competing interests: The authors declare that no competing interests exist.

## Introduction

Regulation of gene expression is fundamental for adaptation to environmental changes in all types of cells. Therefore, gene expression is strictly regulated at all stages from transcription to translation. Regulation of meiotic gene expression is especially crucial to differentiate between the somatic and germ cells. It has been demonstrated that ectopic expression of germline genes causes various cellular defects, including genome instability and tumorigenesis (*Folco et al., 2017*; *Greve et al., 2015*; *Janic et al., 2010*).

Hundreds of genes are upregulated when the fission yeast *Schizosaccharomyces pombe* cells enter meiosis from the mitotic cell cycle in response to nutrient starvation (*Mata et al., 2002*). During the mitotic cell cycle, meiotic genes are strictly suppressed by post-transcriptional mechanisms, in addition to transcriptional regulation, since mistimed expression of meiotic genes severely impairs cell growth. A large number of meiosis-specific transcripts carry a specific region called DSR (determinant of selective removal) and are recognized by a YTH family RNA-binding protein, Mmi1, in mitotically growing *S. pombe* cells. Mmi1 then induces nuclear exosome-mediated RNA elimination (*Harigaya et al., 2006*; *Yamanaka et al., 2010*). DSR activity is exhibited by enriched repeats of the hexanucleotide UNAAAC motif (*Hiriart et al., 2012*; *Yamashita et al., 2012*). The Mmi1 YTH domain preferentially binds to the unmethylated UNAAAC motif, contrasting with the YTH domains in other organisms including mammals, which selectively bind to $N^6$-methyladenosine-containing RNAs (*Chatterjee et al., 2016*; *Wang et al., 2016*; *Wu et al., 2017*). The DSR region has been

found in a group of meiotic transcripts including *mei4,* which encodes a key meiotic transcription factor (*Horie et al., 1998*), and *ssm4,* which encodes a subunit of the dynactin complex (*Niccoli et al., 2004*). Red1, a zinc-finger protein, is another crucial factor involved in the Mmi1-driven RNA elimination (*Sugiyama and Sugioka-Sugiyama, 2011*; *Yamashita et al., 2013*). Red1 constitutes a complex termed MTREC (Mtl1-Red1 core) or NURS (nuclear RNA silencing) with the Mtr4-like RNA helicase, Mtl1, and transfers the Mmi1-bound meiotic transcripts to the nuclear exosome (*Egan et al., 2014*; *Lee et al., 2013*; *Zhou et al., 2015*). In human cells, a similar protein complex, PAXT, composed of a Red1-related zinc-finger protein (ZFC3H1) and an Mtr4 ortholog (hMTR4), has been reported to induce nuclear exosome-dependent RNA degradation (*Meola et al., 2016*). Recently, ZFC3H1 and hMtr4 have also been shown to prevent nuclear export of non-coding RNAs (*Ogami et al., 2017*).

Mmi1 forms several dot structures in the nucleus of the mitotically growing cells (*Harigaya et al., 2006*). Many factors cooperating with Mmi1, including Red1 and exosome subunits, localize to the Mmi1 foci (*Sugiyama and Sugioka-Sugiyama, 2011*; *Yamanaka et al., 2010*; *Yamashita et al., 2013*), suggesting that the foci are the site of degradation of the DSR-containing meiotic transcripts; however, the precise location of the Mmi1 foci in the nucleus remains elusive.

When *S. pombe* cells initiate meiosis, Mmi1-mediated RNA degradation must be suppressed so that DSR-containing meiotic transcripts are expressed. Downregulation of Mmi1 during meiosis is achieved by sequestration of Mmi1 to a meiosis-specific chromosome-associated dot, Mei2 dot (*Harigaya et al., 2006*). Mei2 dot is composed of an RNA-binding protein, Mei2, and its binding partner, meiRNA, and is located at the *sme2* locus, which encodes meiRNA (*Shimada et al., 2003*; *Watanabe et al., 1997*; *Yamashita et al., 1998*). meiRNA carries the DSR region and is targeted by Mmi1, suggesting that meiRNA serves as a decoy for luring Mmi1 (*Shichino et al., 2014*; *Yamashita et al., 2012*).

Mmi1 also plays vital roles in a wide range of regulation of target gene expression, in addition to RNA elimination. Mmi1 induces the assembly of facultative heterochromatin at a subset of its target genes (*Hiriart et al., 2012*; *Tashiro et al., 2013*; *Zofall et al., 2012*). Binding of Mmi1 to target transcripts triggers the recruitment of factors involved in heterochromatin formation along with Red1. It has been demonstrated that a highly conserved multiprotein complex, Ccr4/Not, participates in the Mmi1-mediated heterochromatin formation, albeit to a limited extent (*Cotobal et al., 2015*; *Sugiyama et al., 2016*). Mmi1 has also been shown to accelerate deadenylation by Ccr4/Not, although its in vivo significance remains unknown (*Stowell et al., 2016*). Furthermore, Mmi1 regulates the termination of transcription of its target genes (*Chalamcharla et al., 2015*; *Shah et al., 2014*; *Touat-Todeschini et al., 2017*). Our recent study has revealed that Mmi1-mediated termination of an upstream non-coding RNA ensures the expression of downstream genes (*Touat-Todeschini et al., 2017*).

Mmi1-mediated multilevel regulation is obviously important for proper gene expression in the mitotically growing cells. However, the mechanism of Mmi1-mediated regulation is still not completely known. In this study, we report that Mmi1 prevents the mistimed expression of meiotic proteins by tethering their mRNAs to the nuclear foci. Furthermore, we demonstrated that Mmi1 interacts with itself through the N-terminal region, assisted by a highly conserved cofactor, Erh1. Mmi1 self-interaction is critical for the nuclear retention of meiotic transcripts and consequent prevention of their translation. These results revealed a novel facet of Mmi1-mediated regulation.

## Results

### Mmi1 prevents expression of meiotic proteins in cells with impaired RNA degradation

Disruption of *mmi1* results in a severe growth defect due to the aberrant expression of meiotic transcripts. This defect was alleviated by the deletion of *mei4* (*Harigaya et al., 2006*), indicating that *mei4* is a crucial target of Mmi1-mediated elimination. *red1Δ* cells inappropriately accumulate meiotic transcripts such as *mei4* and *ssm4* during vegetative growth, but are viable albeit showing cold and weak high-temperature sensitivity (*Figure 1A,B*, *Figure 1—figure supplement 1A*, and *Figure 1—figure supplement 2A,B*) (*Sugiyama and Sugioka-Sugiyama, 2011*; *Yamashita et al., 2013*). Deletion of *mei4* did not suppress this temperature-sensitivity (*Figure 1A*), suggesting that these defects are independent of meiotic transcript accumulation. This was also the case in a

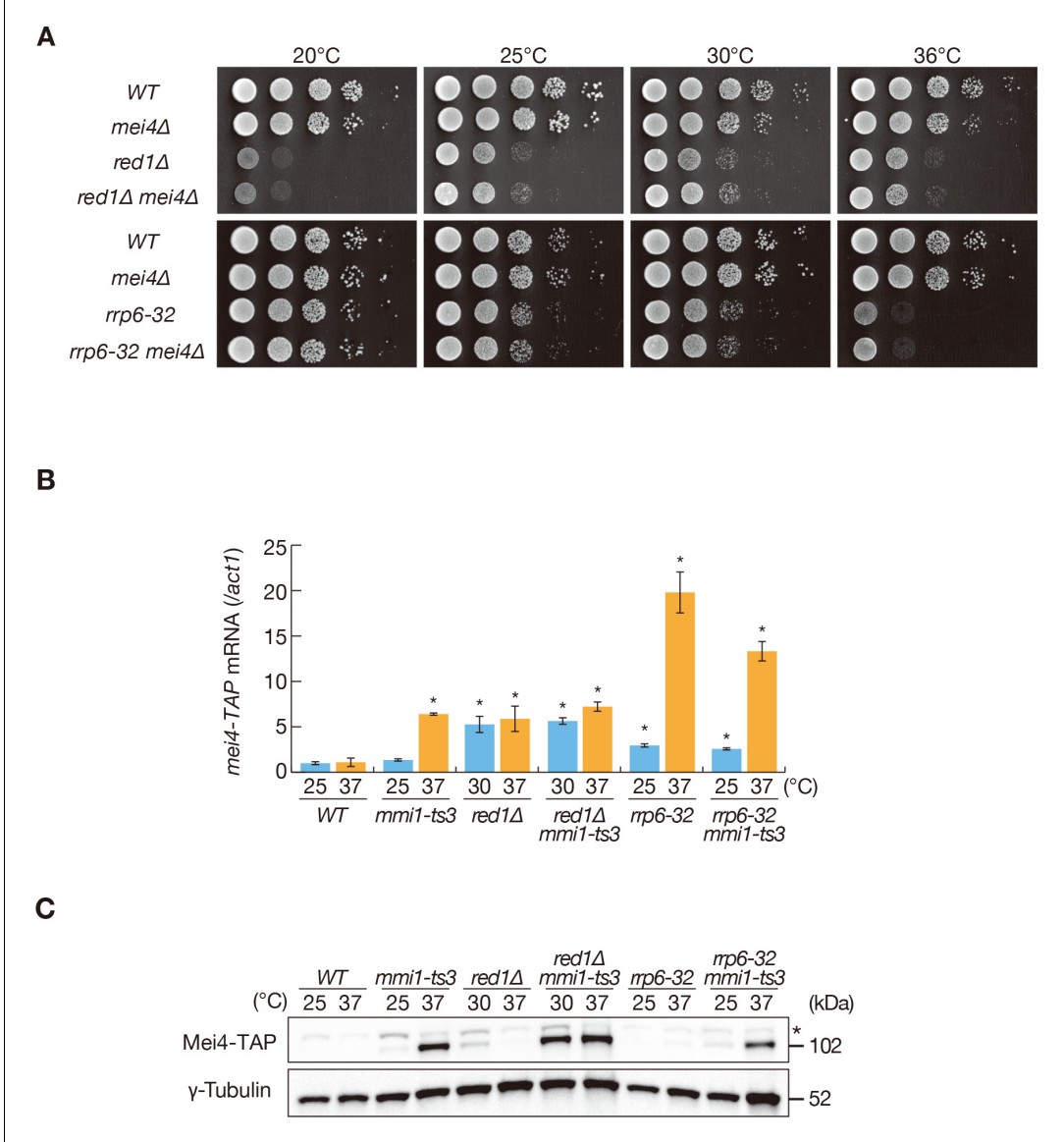

**Figure 1.** Mmi1 prevents expression of proteins encoded by ectopically accumulated meiotic transcripts. (**A**) Growth profiles of *red1Δ mei4Δ* and *rrp6-32 mei4Δ* double mutant cells. 10-fold serial dilutions of wild-type (*WT*), *mei4Δ*, *red1Δ*, *red1Δ mei4Δ*, *rrp6-32*, and *rrp6-32 mei4Δ* cells were spotted on YE media and incubated at the indicated temperatures. (**B**) Expression of *mei4-TAP* mRNAs in *WT*, *mmi1-ts3*, *red1Δ*, *red1Δ mmi1-ts3*, *rrp6-32*, and *rrp6-32 mmi1-ts3* strains. Cells of *WT*, *mmi1-ts3*, *rrp6-32*, and *rrp6-32 mmi1-ts3* expressing *mei4-TAP* from the endogenous locus were grown at 25°C and shifted to 37°C for 4 hr. Cells of *red1Δ* and *red1Δ mmi1-ts3* were grown at 30°C and shifted to 37°C for 4 hr; this was done because *red1Δ* cells display cold sensitivity. *mei4* transcripts were quantified by quantitative RT-PCR and normalized to *act1* encoding actin. Error bars represent standard error from three independent samples. *p<0.05 compared with the wild-type strain at 37°C (Student's *t*-test). (**C**) Expression levels of Mei4-TAP protein in the same conditions as (**B**). γ-tubulin was used as a loading control. The asterisk indicates non-specific bands.

DOI: https://doi.org/10.7554/eLife.32155.002

The following source data and figure supplements are available for figure 1:

**Source data 1.** Source data relating to *Figure 1B* and *Figure 1—figure supplement 1A*.

DOI: https://doi.org/10.7554/eLife.32155.005

**Figure supplement 1.** Expression of Ssm4 protein is prevented by Mmi1.

DOI: https://doi.org/10.7554/eLife.32155.003

**Figure supplement 2.** DSR-containing meiotic transcripts are accumulated when Mmi1-mediated RNA degradation is impaired.

DOI: https://doi.org/10.7554/eLife.32155.004

temperature-sensitive mutant of *rrp6*, which encodes a catalytic subunit of the nuclear exosome (*Figure 1A*). To analyze the discrepancy between growth profiles of *mmi1* and *red1* mutants, we investigated the expression of transcripts and proteins of *mei4* in temperature-sensitive *mmi1-ts3* and *red1Δ* cells. *mei4* transcripts were ectopically accumulated in both mutants during the mitotic cell cycle (*Figure 1B* and *Figure 1—figure supplement 2A*). Strikingly, Mei4 protein expression was hardly detected in *red1Δ* cells, but accumulated in cells with reduced Mmi1 activity (*Figure 1C*). In the temperature-sensitive *rrp6-32* mutant, expression of Mei4 protein was suppressed, whereas *mei4* transcripts accumulated, as seen in *red1Δ* cells (*Figure 1B,C* and *Figure 1—figure supplement 2A*). Mei4 protein was detected in both *mmi1-ts3 red1Δ* and *mmi1-ts3 rrp6-32* double mutants, indicating that Mei4 protein expression requires both transcript accumulation and Mmi1 inactivation. Similarly, protein expression of *ssm4*, another DSR-containing gene, was detected when Mmi1 activity was dampened (*Figure 1—figure supplement 1B*). These results suggest that Mmi1 likely inhibits the expression of proteins encoded by ectopically accumulated DSR-containing mRNAs.

Mmi1 is shown to regulate the transcription termination of its targets (*Chalamcharla et al., 2015*; *Shah et al., 2014*; *Touat-Todeschini et al., 2017*). Northern blot analysis revealed that *mei4* and *ssm4* transcripts in *mmi1-ts3* cells were similar in size to the wild-type meiotic counterparts (*Figure 1—figure supplement 2C,D*), suggesting that Mmi1 has little, if any, impact on the termination of these genes. In *red1Δ* or *rrp6-32* cells, a slight increase in the size of both *mei4* and *ssm4* transcripts was observed (*Figure 1—figure supplement 2A,B*), thereby suggesting a role of these factors in the regulation of termination, including polyadenylation. In cells with impaired Red1 or Rrp6 function, longer *ssm4* transcripts accumulated (*Figure 1—figure supplement 2B*), which were seen to comprise of *ssm4* and its upstream gene (*Zofall et al., 2012*).

## Mmi1 inhibits nuclear export of meiotic transcripts

Since Mmi1 specifically interacts with DSR-containing transcripts and localizes to nuclear foci (*Harigaya et al., 2006*; *Yamashita et al., 2012*), we hypothesized that Mmi1 may capture DSR-containing transcripts in the nuclear foci and sequester them from the translation machinery, even when selective RNA degradation is impaired. We examined the cellular distribution of *mei4* and *ssm4* transcripts using single-molecule RNA fluorescence in situ hybridization (smFISH). We detected a concentration of *mei4* and *ssm4* transcripts in the nucleus in *red1Δ* or *rrp6-32* cells (*Figure 2A,B* and *Figure 2—figure supplement 1A,B*). In *red1Δ* and *rrp6-32* cells, most nuclear RNA spots were larger than cytoplasmic spots, suggesting that multiple molecules of *mei4* or *ssm4* transcripts could have converged in the nucleus. Because we counted each such large spot as one, the number of nuclear spots may possibly be underestimated in *red1Δ* and *rrp6-32* cells. In contrast, when the Mmi1 function was compromised, these transcripts were exported from the nucleus. These observations support the idea that Mmi1 impedes the translation of DSR-containing meiotic transcripts by retaining them in the nucleus. The cytoplasmic distribution of *mei4* and *ssm4* transcripts in *mmi1* mutant cells was halted by the inactivation of Rae1 (*Figure 2C,D* and *Figure 2—figure supplement 1C,D*), which is an essential factor in the nuclear export of mRNAs (*Brown et al., 1995*; *Yoon et al., 1997*). This suggests that the nuclear export of DSR transcripts is mediated by the canonical mRNA export pathway. Elevation of *mei4* and *ssm4* transcript levels was observed in *rae1-167* cells, as shown previously (*Sugiyama et al., 2013*).

## DSR-containing transcripts form Mmi1-dependent intranuclear foci

To gain further insight into the mechanism underlying the nuclear retention of DSR-containing transcripts, we visualized the localization of reporter transcripts (comprised of U1A tag, the firefly luciferase open reading frame, and 0–24 copies of the DSR motif) by coexpressing U1A-YFP (*Figure 3A*). We previously demonstrated that more than 5 copies of the DSR motif exert DSR activity (*Yamashita et al., 2012*). Transcripts carrying eight and more DSR motifs in our reporter system were degraded and decay fragments containing U1A tag were observed, whereas transcripts carrying four motifs were stably expressed (*Figure 3—figure supplement 1A*). We detected foci in mitotic cells expressing the reporter constructs with eight and more DSR motifs (*Figure 3B,C* and *Figure 3—figure supplement 1B*). These findings indicate that the observed foci may have formed from the degraded fragments of reporter transcripts carrying U1A tag. Foci formation took place inside the nucleus, confirmed by simultaneous observation of the nucleoporin Nup60 (*Figure 3D*).

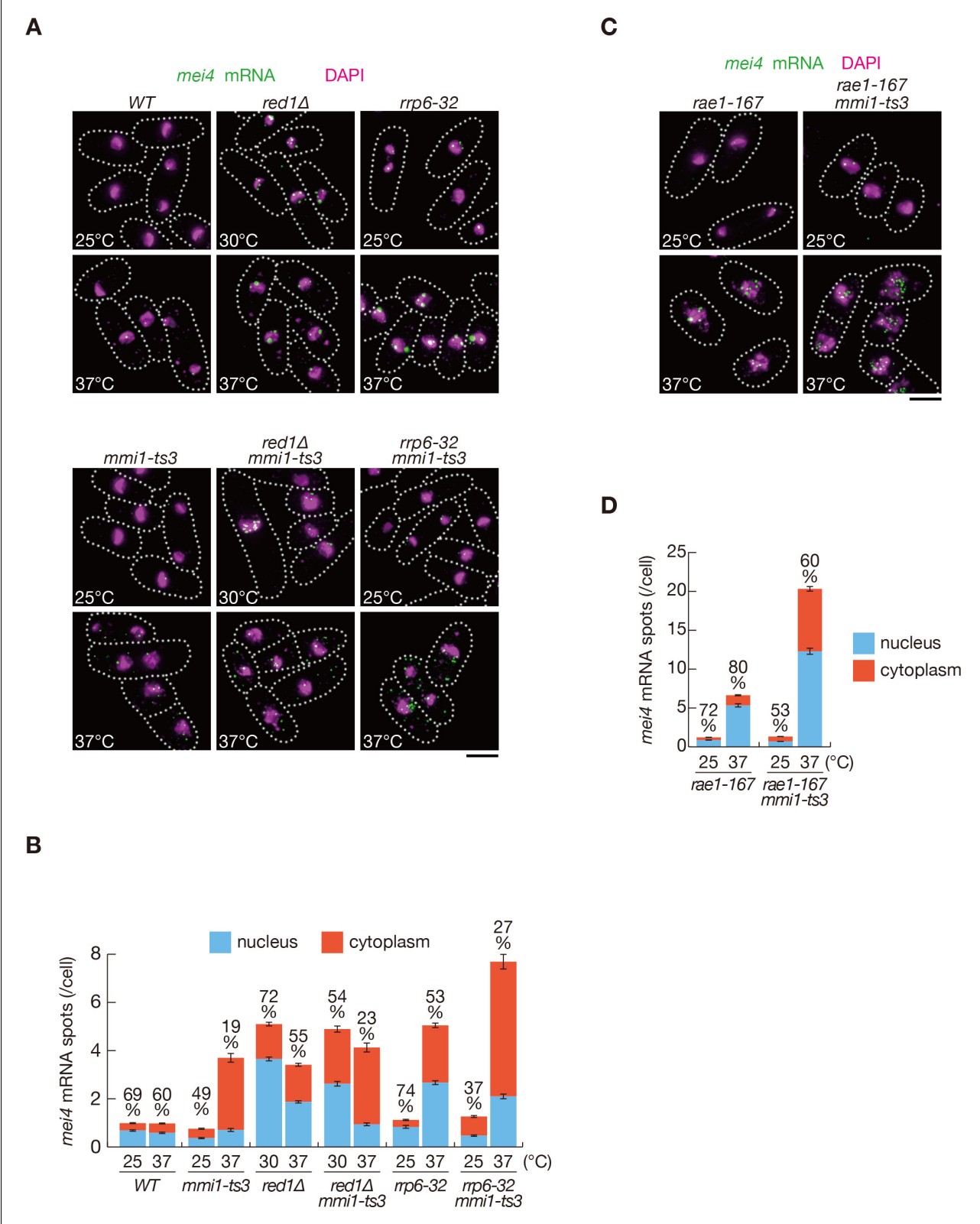

**Figure 2.** Meiotic mRNAs are accumulated in the nucleus depending on Mmi1. (A) Localization of *mei4* mRNAs in wild-type (*WT*), *red1Δ*, *rrp6-32*, *mmi1-ts3*, *red1Δ mmi1-ts3*, and *rrp6-32 mmi1-ts3* strains. Cells of *WT*, *rrp6-32*, *mmi1-ts3*, and *rrp6-32 mmi1-ts3* were grown at 25°C and shifted to 37°C for 4 hr. Cells of *red1Δ* and *red1Δ mmi1-ts3* were grown at 30°C and shifted to 37°C for 4 hr. Single-molecule FISH detected *mei4* mRNAs (green). Nuclear DNA was stained with DAPI (magenta). Dotted lines indicate the shape of cells. Scale bar: 5 μm. (B) Quantification of *mei4* mRNA localization
*Figure 2 continued on next page*

*Figure 2 continued*

in cells shown in (**A**). Number of nuclear (blue) and cytoplasmic (red) signals from each cell was measured. The mean ±standard error of mean from more than 100 cells is shown. Percentages of *mei4* mRNAs in the nucleus are shown above. (**C**) Localization of *mei4* mRNAs in *rae1-167* and *rae1-167 mmi1-ts3* strains. Cells were grown at 25°C and shifted to 37°C for 4 hr. Single-molecule FISH detected *mei4* mRNAs (green). Nuclear DNA was stained with DAPI (magenta). Dotted lines indicate the shape of cells. Scale bar: 5 µm. (**D**) Quantification of *mei4* mRNA localization in cells shown in (**C**). Number of nuclear (blue) and cytoplasmic (red) signals of each cell was measured. The mean ±standard error of mean from more than 100 cells is shown. Percentages of *mei4* mRNAs in the nucleus are shown above.

DOI: https://doi.org/10.7554/eLife.32155.006

The following source data and figure supplement are available for figure 2:

**Source data 1.** Source data relating to *Figure 2B,D* and *Figure 2—figure supplement 1B,D*.
DOI: https://doi.org/10.7554/eLife.32155.008
**Figure supplement 1.** *ssm4* mRNAs are accumulated in the nucleus, dependent on Mmi1.
DOI: https://doi.org/10.7554/eLife.32155.007

Foci were scarcely observed in cells expressing transcripts containing 0 and 4 DSR motifs, suggesting that, unlike smFISH, transcripts could be detected in this system only when a certain concentration of molecules were gathered. The frequency of cells exhibiting foci and the number of foci per cell increased with the number of DSR motifs (*Figure 3C*). DSR-transcript foci coincided with Mmi1 (*Figure 3E*). Foci formation was severely impaired in *mmi1Δ* cells, although the transcripts escaped degradation and accumulated (*Figure 3F,G* and *Figure 3—figure supplement 2A,B,C*). These observations suggest that DSR transcripts were dispersed in the absence of Mmi1. In *red1Δ* and *rrp6-32* cells, in which degradation of DSR transcripts was compromised, foci became prominent, and the frequencies of cells carrying four and more foci increased (*Figure 3F,G* and *Figure 3—figure supplement 2A,B,C*). These foci also colocalized with Mmi1 inside the nucleus (*Figure 3—figure supplement 2D,E*).

## Mmi1 interacts with itself through an N-terminal self-interaction domain

Next, we investigated the mechanism by which Mmi1 forms nuclear foci. Mmi1 lacking the C-terminal YTH domain (Mmi1-ΔYTH) showed similar localization to that of full-length Mmi1 (*Figure 4A,B*). We then attempted to determine the region responsible for foci formation in the N-terminus of Mmi1, and found that the domain of residues 61–180 had the ability to form nuclear foci (*Figure 4B* and *Figure 4—figure supplement 1A*). Two-hybrid analyses and immunoprecipitation assays indicated that Mmi1 interacted with itself through this domain (*Figure 4C,D* and *Figure 4—figure supplement 1B,C*), so we designated this region SID (self-interaction domain). Mmi1 lacking SID (Mmi1-ΔSID) retained the ability to bind to the DSR region (*Figure 4—figure supplement 1D*). However, Mmi1-ΔSID lost the function to induce the elimination of DSR-containing transcripts (*Figure 4E* and *Figure 4—figure supplement 1E*) and could not rescue the growth defect of temperature-sensitive *mmi1* mutant cells (*Figure 4F*). It was also noted that expression of Mmi1-ΔYTH had a dominant-negative effect (*Figure 4E,F* and *Figure 4—figure supplement 1E*). Deletion of SID also resulted in loss of interaction with Red1 (*Figure 4G*), suggesting that Mmi1 interacts with factors involved in Mmi1-mediated elimination through SID.

## Self-interaction domain is required for nuclear retention of meiotic transcripts and prevention of their protein expression

Next, we assessed whether removing SID from Mmi1 affects the nuclear retention and expression of DSR-containing transcripts. *mei4* and *ssm4* transcripts were mainly observed in the cytoplasm of *mmi1* mutant cells expressing Mmi1-ΔSID, as cells expressing Mmi1-ΔYTH (*Figure 5A,B* and *Figure 5—figure supplement 1A,B*). Nuclear foci of the reporter transcripts were dispersed in cells expressing Mmi1-ΔSID (*Figure 5—figure supplement 2A,B*), although full-length transcripts accumulated (*Figure 5—figure supplement 2C*). Correspondingly, both mRNAs and proteins of *mei4* and *ssm4* were expressed strongly in *mmi1-ts3* cells expressing Mmi1-ΔSID (*Figure 4E*, *Figure 4—figure supplement 1E*, *Figure 5C* and *Figure 5—figure supplement 3*). In *mmi1-ts3* cells expressing Mmi1-ΔYTH, Mei4 protein was expressed at a permissive temperature, confirming the dominant-negative effect of Mmi1-ΔYTH (*Figure 5C*). These observations indicate that SID is crucial for tethering DSR-containing meiotic transcripts to nuclear foci.

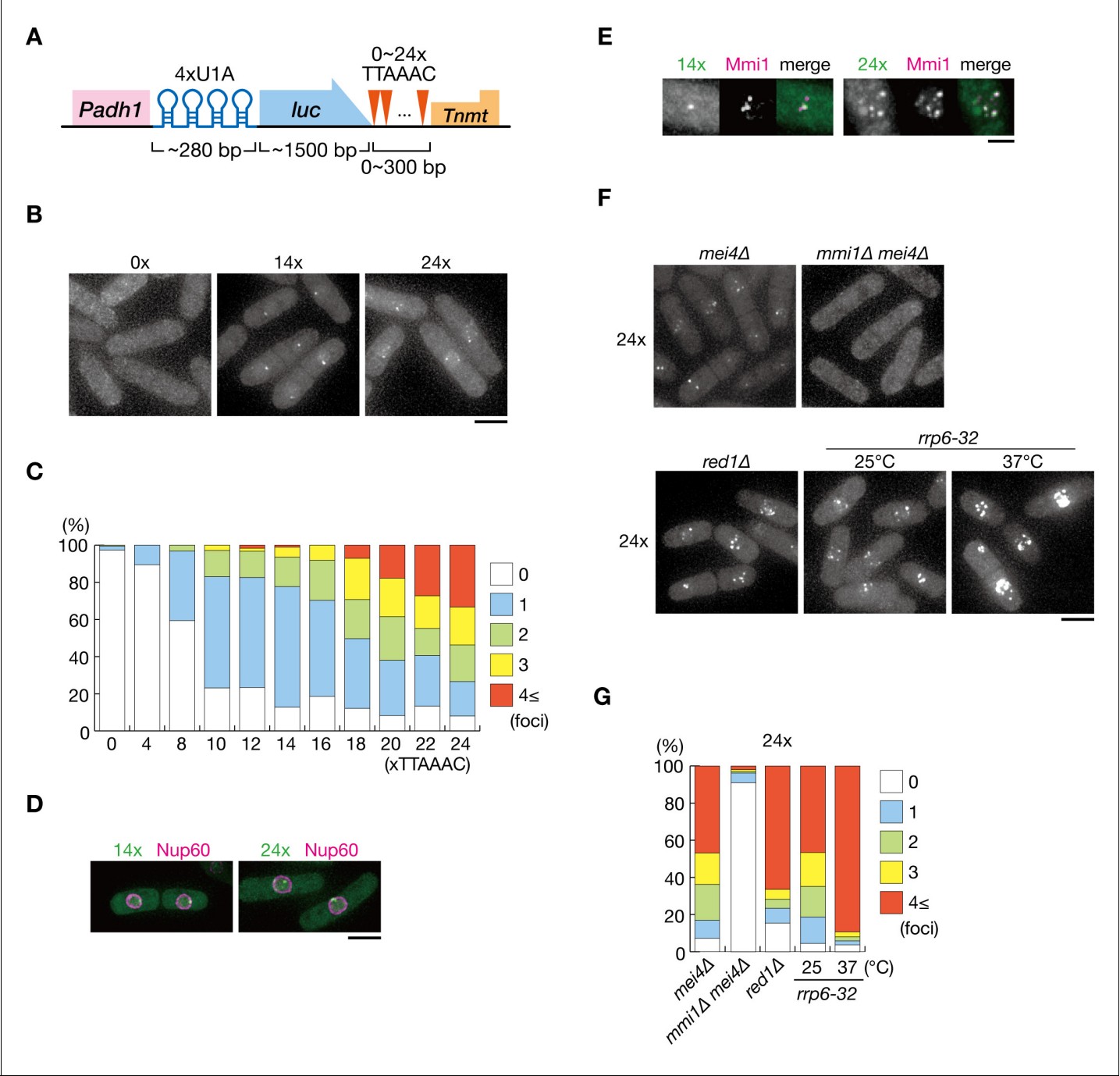

**Figure 3.** DSR-containing transcripts form Mmi1-dependent nuclear foci. (**A**) Schematic of the DSR-containing reporter gene composed of the *adh1* promoter, 4x U1A tag, the luciferase ORF, 0–24 copies of the DSR motif (TTAAAC), and the *nmt1* terminator. RNA was visualized by coexpressing U1A-YFP. (**B**) Localization of reporter transcripts carrying 0, 14, and 24 copies of the DSR motif in wild-type cells. Scale bar: 5 μm. (**C**) Percentages of cells containing 1, 2, 3, or 4 and more reporter transcripts foci (n > 100). (**D**) Localization of reporter transcripts carrying 14 and 24 copies of the DSR motif (green) and the nucleoporin Nup60 (magenta) in wild-type cells. Scale bar: 5 μm. (**E**) Localization of reporter transcripts with 14 and 24 copies of the DSR motif (green) and Mmi1 (magenta) in wild-type cells. Images of the nuclear region are shown. Scale bar: 2 μm. (**F**) Localization of reporter transcripts with 24 copies of the DSR motif in *mei4Δ*, *mmi1Δ mei4Δ*, *red1Δ*, and *rrp6-32* strains. *mei4Δ*, *mmi1Δ mei4Δ*, and *red1Δ* cells were grown at 30°C. *rrp6-32* cells were grown at 25°C and then shifted to 37°C for 4 hr. Scale bar: 5 μm. (**G**) Percentages of cells containing 1, 2, 3, or 4 and more reporter transcript foci with 24 copies in *mei4Δ*, *mmi1Δ mei4Δ*, *red1Δ*, and *rrp6-32* strains (n > 100).

DOI: https://doi.org/10.7554/eLife.32155.009

The following source data and figure supplements are available for figure 3:

*Figure 3 continued on next page*

*Figure 3 continued*

**Source data 1.** Source data relating to *Figure 3C,G*, and *Figure 3—figure supplement 2B*.
DOI: https://doi.org/10.7554/eLife.32155.012
**Figure supplement 1.** DSR-containing transcripts form intranuclear foci.
DOI: https://doi.org/10.7554/eLife.32155.010
**Figure supplement 2.** Foci formation of DSR transcripts is dependent on Mmi1, but not on Red1 or Rrp6.
DOI: https://doi.org/10.7554/eLife.32155.011

## Mmi1 self-interaction is assisted by Erh1

To identify factors involved in the foci formation of Mmi1, we next examined the localization of Mmi1 in deletion mutants lacking genes encoding factors related to Mmi1-mediated elimination, including Red1 (*Sugiyama and Sugioka-Sugiyama, 2011*; *Sugiyama et al., 2012*; *Yamashita et al., 2013*). The deletion of *red1*, *rhn1/iss4*, and *iss9* had no impact on Mmi1 foci formation (*Figure 6—figure supplement 1A*). Rhn1 and Iss9 also did not colocalize with Mmi1 (*Figure 6—figure supplement 1B*). In cells lacking *erh1*, which encodes a homolog of Enhancer of Rudimentary (*Krzyzanowski et al., 2012*; *Yamashita et al., 2013*), the frequency of cells carrying Mmi1 foci was decreased (*Figure 6A*). Erh1 colocalized and interacted with Mmi1 (*Figure 6B,C*), as shown previously (*Sugiyama et al., 2016*). The interaction between Erh1 and Mmi1 was impaired when Mmi1 lacked SID (*Figure 6C*). Furthermore, Mmi1 self-interaction was severely dampened, although not completely abrogated, in *erh1Δ* cells (*Figure 6D*) while the deletion of *erh1* had no effect on the interaction between Mmi1 and Red1 (*Figure 6—figure supplement 1C*). These evidences suggest that Erh1 might be engaged in Mmi1-mediated elimination, by binding to SID and reinforcing the self-interaction and foci formation of Mmi1, although the self-interaction may be induced by Mmi1 itself. This is consistent with the two-hybrid analysis described above (*Figure 4C* and *Figure 4—figure supplement 1B,C*).

Next, we examined the distribution of DSR-containing transcripts in *erh1Δ* cells. *mei4* and *ssm4* transcripts were exported to the cytoplasm, as cells lacking SID (*Figure 5A,B* and *Figure 5—figure supplement 1A,B*). In *erh1Δ* cells, nuclear foci formation of the reporter transcripts was also severely impaired (*Figure 5—figure supplement 2A,B*). Transcripts and proteins of *mei4* and *ssm4* were accumulated in *erh1Δ* cells, although to a lesser extent than in *mmi1-ts3* cells (*Figure 6E,F* and *Figure 6—figure supplement 1D,E*). The *erh1Δ* cells displayed cold sensitivity (*Sugiyama et al., 2016*; *Yamashita et al., 2013*), and this growth defect was suppressed by the deletion of *mei4* (*Figure 6G*), as was the case in *mmi1* deletion. From these observations, we conclude that DSR-containing transcripts are retained at nuclear foci through Mmi1 self-interaction, which is assisted by Erh1.

Recently, we have shown that Mei4 protein is expressed in cells lacking Pab2, a nuclear poly(A)-binding protein, which acts in Mmi1-mediated RNA degradation (*Cotobal et al., 2015*). We compared the expression levels of Mei4 protein in *mmi1-ts3*, *red1Δ*, and *pab2Δ* cells (*Figure 6—figure supplement 2*). In *pab2Δ* cells, Mei4 was expressed very weakly, at a level similar to that in *red1Δ* cells. This indicates that Pab2 does not play a significant role in Mmi1-driven nuclear tethering of DSR-containing transcripts.

## Discussion

Our data demonstrated that Mmi1 controls the expression of meiotic genes in a multilayered fashion, namely through heterochromatin silencing, selective elimination of transcripts, and prevention of protein expression by nuclear foci tethering (*Figure 7*). The meiotic transcripts carrying DSR are recognized by Mmi1, localized in nuclear foci through Mmi1 self-interaction, and degraded by the nuclear exosome. Induction of RNA degradation might be a primary function of Mmi1. However, even though RNA degradation is impaired, nuclear export of the DSR-containing transcripts is prevented by Mmi1 and consequently their protein expression is suppressed. Transcription of some Mmi1-target gene loci, including *mei4,* is also silenced through Mmi1-mediated facultative heterochromatin formation (*Hiriart et al., 2012*; *Tashiro et al., 2013*; *Zofall et al., 2012*). In the *red1* and *rrp6* mutant cells, degradation of meiotic transcripts is dampened. However, transcripts are tethered to the nuclear Mmi1 foci and their expression is suppressed through sequestration from the

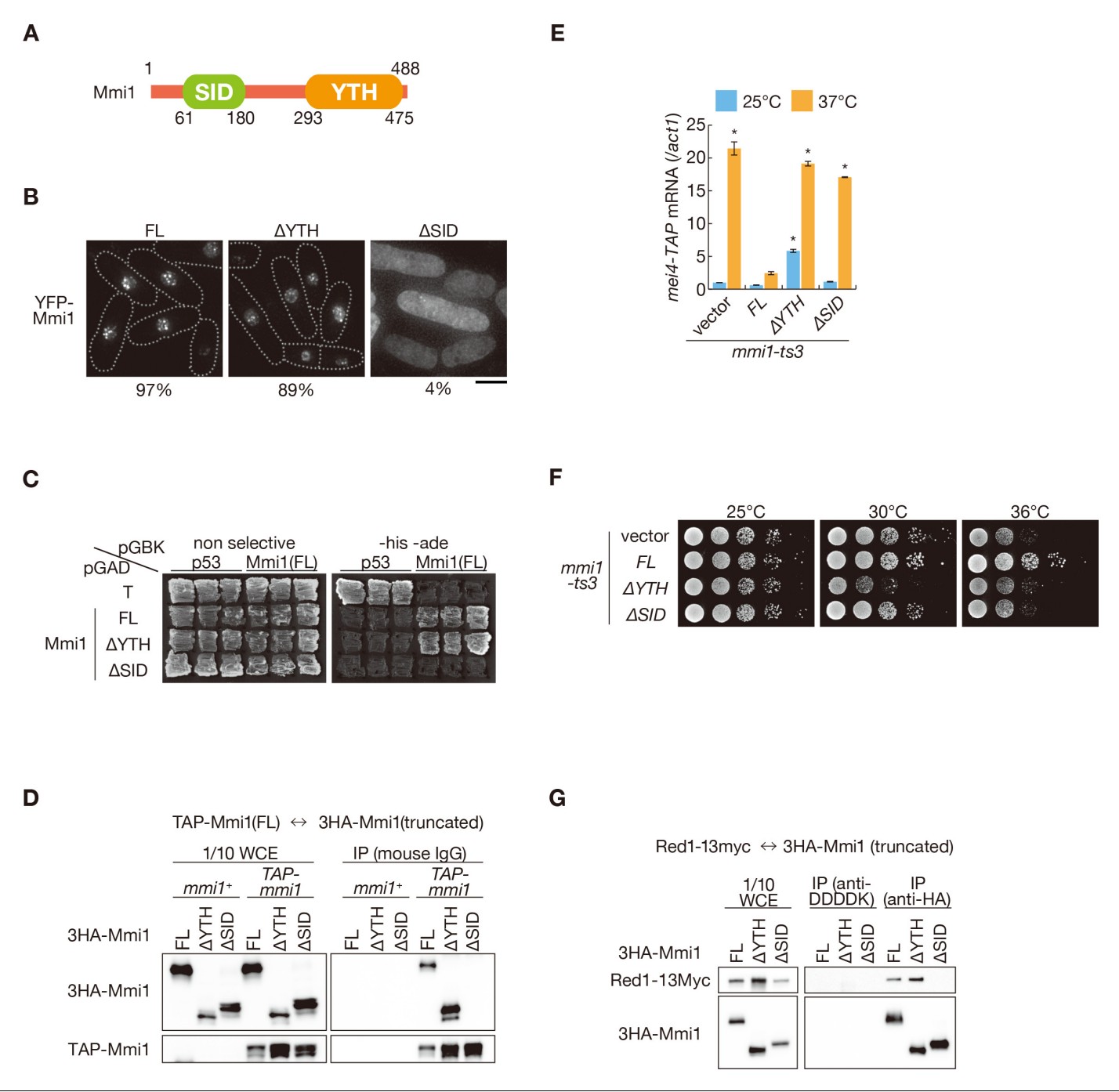

**Figure 4.** Mmi1 interacts with itself through the N-terminal SID region. (A) Schematic of the structure of Mmi1. (B) Localization of full-length Mmi1 and Mmi1 lacking the YTH domain or the SID region. YFP-tagged full-length or truncated Mmi1 was expressed from the plasmid with a mild promoter. Frequencies of cells carrying nuclear Mmi1 foci are indicated below (*n* > 100). Dotted lines indicate the shape of cells. Scale bar: 5 µm. (C) Two-hybrid interaction of full-length Mmi1 with truncated Mmi1. p53 and T antigen were used as controls. (D) Mmi1 self-interaction in wild-type cells. Cell extracts prepared from wild-type cells expressing TAP-tagged full-length Mmi1 and HA-tagged full-length or truncated Mmi1 were subjected to immunoprecipitation. Precipitates and 1/10 cell extracts were then immunoblotted. (E) Expression of *mei4-TAP* mRNAs was analyzed by quantitative RT-PCR in *mmi1-ts3* cells expressing full-length Mmi1, Mmi1-ΔYTH, or Mmi1-ΔSID. *mmi1-ts3* cells were grown at 25°C and then shifted to 37°C for 4 hr. *mei4* transcripts were quantified and normalized to *act1*. Error bars represent standard error from three independent samples. *p<0.05 compared with cells expressing full-length Mmi1 at 37°C (Student's *t*-test). (F) Growth profiles of *mmi1-ts3* cells expressing Mmi1 lacking YTH or lacking SID. 10-fold serial dilutions of *mmi1-ts3* cells expressing full-length Mmi1, Mmi1-ΔYTH, and Mmi1-ΔSID from the plasmid were spotted on MM media and incubated at the indicated temperatures. (G) Interaction of Mmi1-ΔYTH and Mmi1-ΔSID with Red1. Cell extracts prepared from wild-type cells expressing HA-

*Figure 4 continued on next page*

Figure 4 continued

tagged full-length Mmi1, Mmi1-ΔYTH or Mmi1-ΔSID, and Myc-tagged Red1 were subjected to immunoprecipitation with an anti-HA antibody. An anti-DDDDK antibody was used as negative control. Precipitates and 1/10 cell extracts were then immunoblotted.

DOI: https://doi.org/10.7554/eLife.32155.013

The following source data and figure supplement are available for figure 4:

**Source data 1.** Source data relating to *Figure 4E* and *Figure 4—figure supplement 1E*.

DOI: https://doi.org/10.7554/eLife.32155.015

**Figure supplement 1.** The N-terminal SID region is essential for Mmi1 self-interaction.

DOI: https://doi.org/10.7554/eLife.32155.014

translation machinery. In the absence of Mmi1, degradation of the DSR-containing meiotic transcripts is compromised and they are exported to the cytoplasm and translated, leading to the deleterious expression of meiotic genes. Recently, ZFC3H1, a possible counterpart of Red1 in human cells, has been demonstrated to induce nuclear exosome-mediated degradation of long noncoding RNAs, and prevent their nuclear export in cooperation with hMtr4, an ortholog of which forms a complex with Red1 (*Ogami et al., 2017*). hMtr4 has also been shown to induce degradation of mRNAs by inhibiting their nuclear export and recruiting the nuclear exosome (*Fan et al., 2017*). It is an intriguing question whether a target-recognition factor, equivalent to Mmi1, works in human cells, although Red1 and ZFC3H1 act differently in nuclear export of the target transcripts.

Regulation of protein expression plays a critical role during meiosis in other organisms. During oogenesis in *Xenopus laevis*, a small subset of mRNAs, such as mRNAs encoding key regulators for meiotic progression, are translationally activated, whereas other mRNAs are repressed (*Sheets et al., 2017*). Changes in poly(A) tail length are known to be important for this regulation (*Weill et al., 2012*). In addition, it has been suggested that the $N^6$-methyladenosine modification, which is often recognized by the YTH domain, is involved in this translational regulation (*Qi et al., 2016*). In *S. pombe*, ribosome profiling has also shown that translation efficiency of the DSR-containing transcripts is indeed upregulated during meiosis (*Duncan and Mata, 2014*). This is consistent with our findings that Mmi1 modulates gene expression profiles, between mitosis and meiosis, by regulating protein expression through nuclear tethering of meiotic mRNAs, in addition to controlling their stability.

The DSR-containing meiotic transcripts localize in the nuclear Mmi1 foci, where many other factors involved in selective elimination exist (*Sugiyama and Sugioka-Sugiyama, 2011*; *Yamanaka et al., 2010*; *Yamashita et al., 2013*), implying that Mmi1 foci are the major sites of meiotic transcript degradation. The Mmi1 foci are distinct from the *mei4* locus, which encodes a key target of Mmi1 (*Egan et al., 2014*; *Shichino et al., 2014*), while enrichment of Mmi1 at the *mei4* locus has been observed by chromatin immunoprecipitation analyses (*Chalamcharla et al., 2015*; *Tashiro et al., 2013*). It is possible that Mmi1 co-transcriptionally recognizes its target transcripts at their gene loci and Mmi1-transcript complexes move to the nuclear foci where RNA degradation takes place; however, we cannot exclude the possibility of Mmi1-inducing RNA degradation on its target gene loci. Further investigation would be required to determine the site of RNA degradation and the precise localization of the Mmi1 foci.

Mmi1 interacts with itself through the N-terminal SID region with assistance from Erh1. Mmi1 self-interaction is likely to trigger nuclear foci formation, which may be a prerequisite for target transcript degradation and their nuclear tethering. The following observations support this proposal: SID is required for foci formation; Mmi1 lacking SID loses its function of inducing degradation of the target transcripts and preventing their protein expression; *erh1Δ* cells, in which Mmi1 self-interaction is impaired, have less Mmi1 foci and show weakened Mmi1 activity, although the interaction between Mmi1 and Red1 is intact. Mmi1 forms foci exclusively in the nucleus. This may be because Mmi1 foci are formed only when a certain number of Mmi1 molecules are enriched in a limited area. However, the possibility of a nuclear-specific factor(s), essential for Mmi1 foci formation, cannot be completely excluded. Further studies would be needed to fully elucidate the mechanism underlying self-interaction and nuclear foci formation of Mmi1.

Erh1 is highly conserved among eukaryotes. Orthologs of Erh1 have been shown to play roles in various processes, including transcriptional and post-transcriptional regulation (*Weng and Luo, 2013*). It is an intriguing question whether ERH proteins play a similar role as the *S. pombe* Erh1.

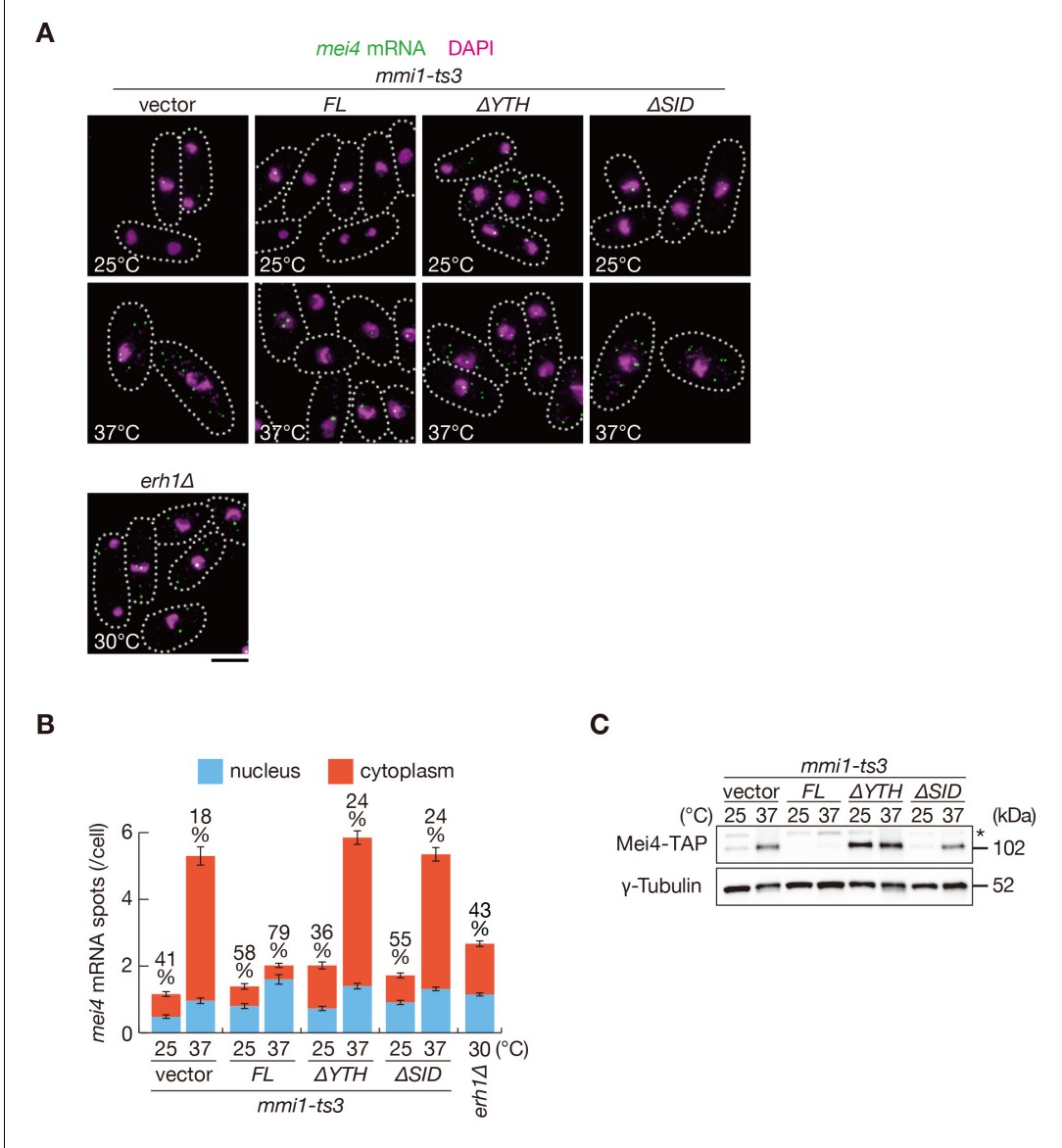

**Figure 5.** Mmi1 self-interaction is crucial for nuclear retention of DSR-containing transcripts and prevention of their protein expression. (**A**) Localization of *mei4* mRNAs in *mmi1-ts3* cells expressing full-length Mmi1, Mmi1-ΔYTH, or Mmi1-ΔSID from the plasmid, and in *erh1Δ* cells. *mmi1-ts3* cells were grown at 25°C and then shifted to 37°C for 4 hr. *erh1Δ* cells were grown at 30°C. Single molecule FISH detected *mei4* mRNAs (green). Nuclear DNA was stained with DAPI (magenta). Dotted lines indicate the shape of cells. Scale bar: 5 μm. (**B**) Quantification of *mei4* mRNA localization in cells shown in (**A**). Number of nuclear (blue) and cytoplasmic (red) signals from each cell was measured. The mean ±standard error of mean from more than 100 cells is shown. Percentages of *mei4* mRNAs in the nucleus are shown above. (**C**) Expression levels of Mei4-TAP protein in *mmi1-ts3* cells expressing full-length Mmi1, Mmi1-ΔYTH, or Mmi1-ΔSID. *mmi1-ts3* cells were grown at 25°C and then shifted to 37°C for 4 hr. γ-Tubulin was used as a loading control. The asterisk indicates non-specific bands.

DOI: https://doi.org/10.7554/eLife.32155.016

The following source data and figure supplements are available for figure 5:

**Source data 1.** Source data relating to *Figure 5B* and *Figure 5—figure supplement 1B*, *2B*.
DOI: https://doi.org/10.7554/eLife.32155.020

**Figure supplement 1.** The N-terminal SID region is required for nuclear retention of *ssm4* mRNAs.
DOI: https://doi.org/10.7554/eLife.32155.017

**Figure supplement 2.** DSR-containing transcripts form nuclear foci through the Mmi1 self-interaction.
DOI: https://doi.org/10.7554/eLife.32155.018

**Figure supplement 3.** Mmi1 self-interaction is crucial for prevention of Ssm4 protein expression.
DOI: https://doi.org/10.7554/eLife.32155.019

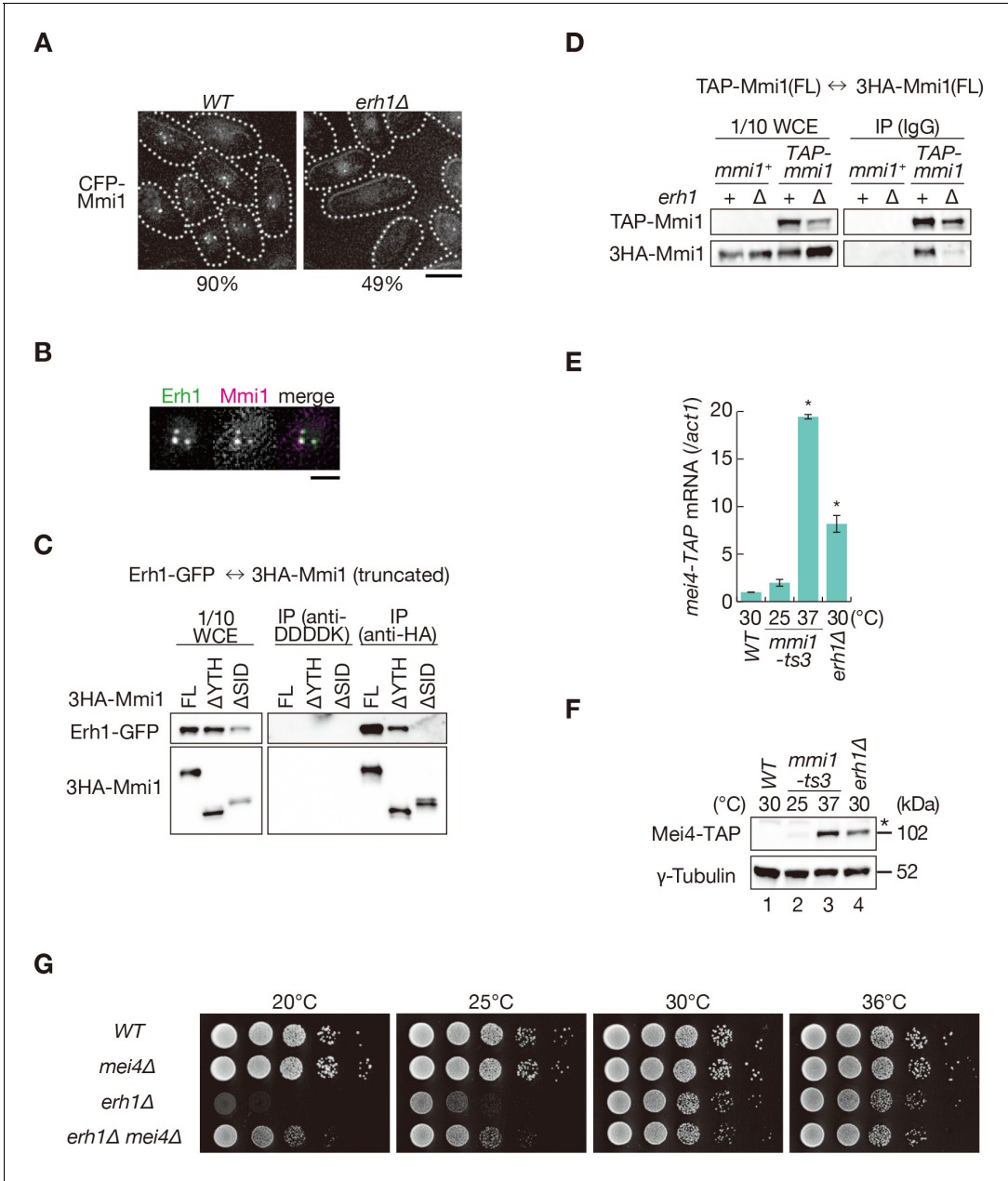

**Figure 6.** Mmi1 interacts with itself with the assistance of Erh1. (**A**) Localization of Mmi1 in *erh1Δ* cells. Frequencies of cells carrying nuclear Mmi1 foci are indicated below (*n* > 100). Dotted lines indicate the shape of cells. Scale bar: 5 μm. (**B**) Localization of YFP-tagged Erh1 expressed from the endogenous locus (green) and CFP-Mmi1 (magenta) in wild-type cells. Images of the nuclear region are shown. Scale bar: 2 μm. (**C**) Interaction of Mmi1-ΔYTH and Mmi1-ΔSID with Erh1. Cell extracts prepared from wild-type cells expressing HA-tagged full-length Mmi1, Mmi1-ΔYTH or Mmi1-ΔSID, and GFP-tagged Erh1 were subjected to immunoprecipitation with an anti-HA antibody. An anti-DDDDK antibody was used as negative control. Precipitates and 1/10 cell extracts were then immunoblotted. (**D**) Mmi1 self-interaction in *erh1Δ* cells. Cell extracts prepared from wild-type and *erh1Δ* cells expressing TAP-tagged Mmi1 and HA-tagged Mmi1 were subjected to immunoprecipitation. Precipitates and 1/10 cell extracts were then immunoblotted. (**E**) Expression of *mei4-TAP* mRNAs was analyzed by quantitative RT-PCR in *erh1Δ* cells. Wild-type (*WT*) and *erh1Δ* cells were grown at 30°C. *mmi1-ts3* cells were grown at 25°C and then shifted to 37°C for 4 hr. *p<0.05 compared with the wild type strain (Student's *t*-test). (**F**) Expression levels of Mei4-TAP protein in the same conditions as (**E**). γ-tubulin was used as a loading control. The asterisk indicates non-specific bands. (**G**) Growth profiles of *erh1Δ mei4Δ* double mutant cells. 10-fold serial dilutions of wild-type, *mei4Δ*, *erh1Δ*, and *erh1Δ mei4Δ* cells were spotted on YE media and incubated at the indicated temperatures.

DOI: https://doi.org/10.7554/eLife.32155.021

The following source data and figure supplements are available for figure 6:

**Source data 1.** Source data relating to *Figure 6E* and *Figure 6—figure supplement 1D*.

*Figure 6 continued on next page*

*Figure 6 continued*

DOI: https://doi.org/10.7554/eLife.32155.024

**Figure supplement 1.** Erh1 assists Mmi1 self-interaction.

DOI: https://doi.org/10.7554/eLife.32155.022

**Figure supplement 2.** Pab2 is not required for prevention of Mei4 protein expression.

DOI: https://doi.org/10.7554/eLife.32155.023

Foci formation by the YTH family proteins is known in cells of higher eukaryotes. YTHDC1/YT521-B, which controls mRNA splicing (*Lence et al., 2016*; *Xiao et al., 2016*), interacts with itself and

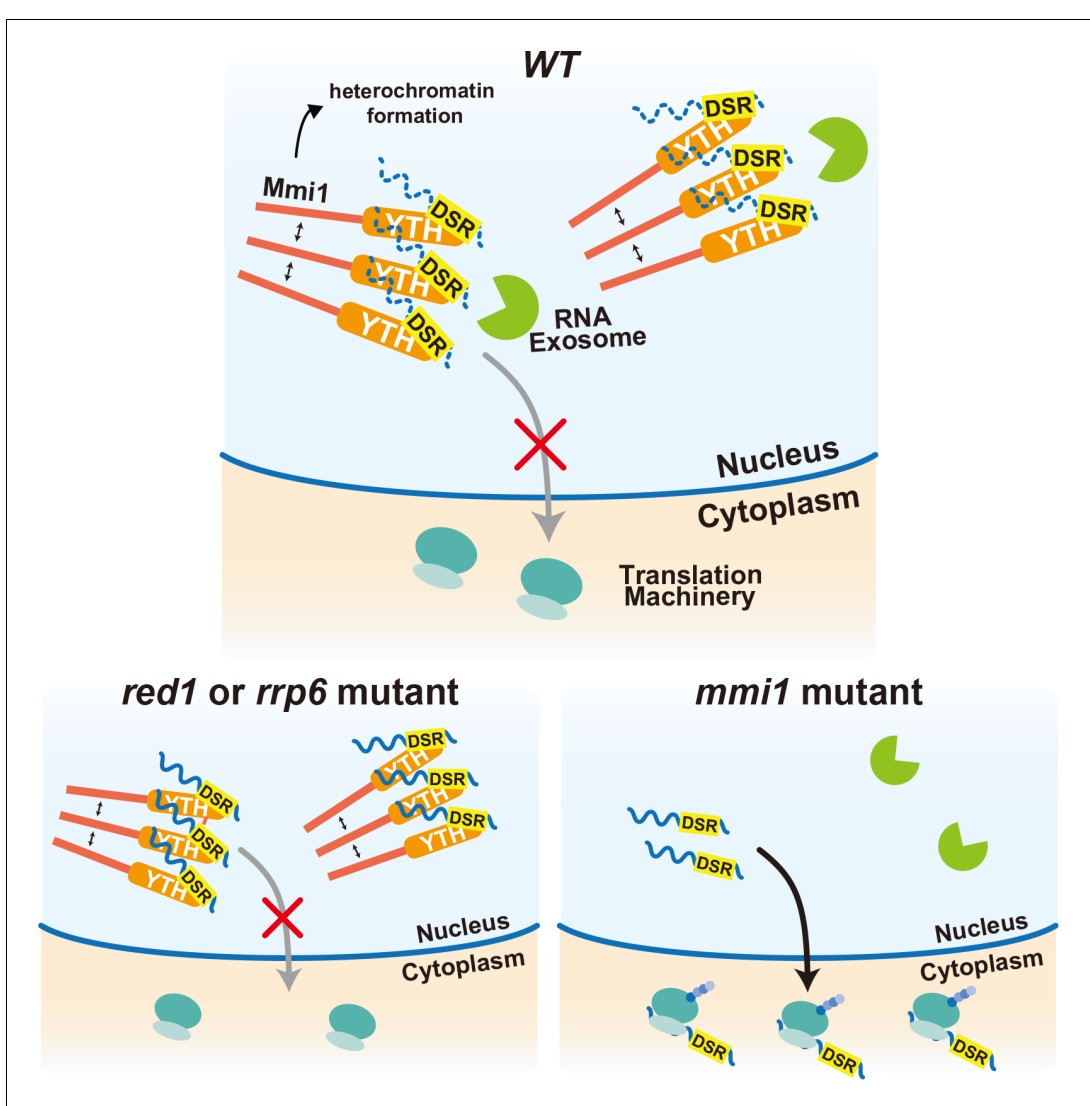

**Figure 7.** Multi-layered regulation to prevent mistimed expression of meiotic genes by the YTH-domain protein Mmi1. In wild-type (*WT*) cells, meiotic transcripts carrying DSR are recognized by Mmi1, localize in nuclear foci through Mmi1 self-interaction, and are degraded by the nuclear exosome. Nuclear export and ectopic protein expression are prevented. Mmi1 also induces heterochromatin formation at its target genes. In *red1* and *rrp6* mutant cells, degradation of meiotic transcripts is dampened. However, transcripts are tethered to nuclear Mmi1 foci and expression of them is suppressed through sequestration from the translation machinery. In the absence of Mmi1, meiotic transcripts carrying DSR are exported to the cytoplasm and translated, leading to the deleterious expression of meiotic genes.

DOI: https://doi.org/10.7554/eLife.32155.025

forms nuclear foci called YT bodies (*Hartmann et al., 1999*; *Nayler et al., 2000*). The cytoplasmic YTH protein YTHDF2, which regulates the stability of transcripts (*Wang et al., 2014*), also aggregates into the cytoplasmic foci and interacts with the processing body, which is enriched in the factors involved in RNA degradation (*Wang et al., 2014*). These YTH-containing proteins have regions that are enriched in particular amino acids (proline, arginine, and glutamic acid in YTHDC1; proline, glutamine, and asparagine in YTHDF2) and these regions are crucial for proper localization of proteins (*Hartmann et al., 1999*; *Wang et al., 2014*). The SID region of Mmi1 is also enriched in serine (20/120 amino acids), arginine (12/120), tyrosine (12/120), and proline (15/120). Most YTH-family proteins do not share sequence similarity without the YTH domain. It is possible that some of them share structural features, such as low complexity regions, and have the ability to form foci through those regions.

Rigorous Mmi1-mediated regulation may function to remove meiotic factors harmful for mitotic growth. Growth retardation of *mmi1* mutant cells is mainly caused by inappropriate accumulation of the Mei4 protein during the mitotic cell cycle. It has also been reported that the ectopic expression of meiotic cohesin *rec8*, induced by the deletion of *mmi1*, causes abnormality in mitotic chromosome segregation (*Folco et al., 2017*). In higher eukaryotes, certain genes are exclusively expressed in germ lines and cancerous tissues. Such genes, named cancer/testis antigen genes, include the meiotic cohesin genes, counterparts of which are targeted by Mmi1 in *S. pombe* (*Feichtinger et al., 2012*; *Simpson et al., 2005*). It would be interesting and important to investigate whether similar mechanisms of Mmi1-mediated regulation operate in other organisms.

## Materials and methods

### Yeast strains, genetic methods, and growth media

The *S. pombe* strains used in this study are listed in *Supplementary file 1*. Complete medium YE, minimal medium SD and MM (*Moreno et al., 1991*), synthetic sporulation medium SSA (*Egel and Egel-Mitani, 1974*), and sporulation agar SPA (*Gutz et al., 1974*) were used. The general genetic methods used to analyze the *S. pombe* strains have been previously described (*Gutz et al., 1974*). Standard gene-targeting protocols were carried out to create deletion mutants and epitope-tagged strains (*Bähler et al., 1998*; *Sato et al., 2005*). Truncated alleles of *mmi1* were constructed using the PrimeSTAR Mutagenesis Basal Kit (Takara, Japan). The *co2* region, a gene-free region on chromosome I, was selected as a target site for chromosomal integration of cloned constructs (*Kakui et al., 2015*).

### RNA preparation and northern blot analysis

*S. pombe* cells were lysed by vortexing vigorously with glass beads in a buffer (0.2 M Tris-HCl [pH7.5], 0.5 M NaCl, 0.01 M ethylenediaminetetraacetic acid (EDTA), 1% sodium dodecyl sulfate (SDS)). After two rounds of extraction with phenol:chloroform:isoamyl alcohol (25:24:1), RNA was recovered by ethanol precipitation. Ten µg of total RNA was denatured with formamide, separated by electrophoresis on a 1.2% agarose gel containing 1% formaldehyde, and blotted onto a GeneScreen Plus membrane (Perkin Elmer, Waltham, Massachusetts). DNA fragments to probe transcripts were amplified by PCR using primers (*Supplementary file 2*) and labeled with [γ-$^{32}$P]dCTP using MegaPrime DNA Labeling System (GE Healthcare, UK). In *Figure 1—figure supplement 2*, RNA was blotted onto a Hybond N + membrane (GE healthcare). DNA fragments to probe the transcripts were amplified by PCR, using suitable primers (*Supplementary file 2*), and DIG-labeled RNA probes were synthesized using the DIG RNA labeling kit (Roche, Switzerland).

### Quantitative reverse-transcription PCR analysis

Total RNA was treated with DNase (Turbo DNA-free, Ambion, Waltham, Massachusetts) and reverse-transcribed to cDNA by using ReverTra Ace qPCR Master Mix (TOYOBO, Japan). Quantitative PCR was performed using LightCycler96 (Roche) with SYBR Premix Ex Taq II (Tli RNaseH Plus) (Takara). Primers used in this study are listed in *Supplementary file 2*.

## Protein extraction and western blot analysis

Harvested cells were lysed by vortexing vigorously with glass beads in HB buffer (25 mM MOPS (pH7.2), 5 mM ethylene glycol bis-(2-aminoehylether) tetraacetic acid (EGTA) (pH7.2), 15 mM MgCl$_2$, 150 mM KCl, 50 mM beta-glycerophosphate, 15 mM $p$-nitrophenylphosphate, 1 mM dithiothreitol (DTT), 1 mM phenylmethylsulfonyl fluoride (PMSF), 0.1 mM Na$_3$VO$_4$, 0.2% NP-40, protease inhibitor cocktail (Complete Mini EDTA-free, Roche)). The following antibodies were used to detect target proteins: rabbit anti-HA Y11 (Santa Cruz, Dallas, Texas), mouse anti-GFP (Roche), rabbit anti-GFP (Life Technologies, Carlsbad, California), mouse anti-γ-tubulin (Sigma), rabbit peroxidase anti-peroxidase soluble complex antibody (Sigma, St. Louis, Missouri), and mouse anti-Myc 9E10 (Santa Cruz).

## Immunoprecipitation

Native cell extracts were incubated with antibody-conjugated magnetic beads for 1 hr at 4°C. The beads were then washed three times. Mouse IgG beads (Cell Signaling Technology, Danvers, Massachusetts) and mouse anti-HA beads (MBL, Japan) were used to precipitate TAP-tagged Mmi1 and HA-tagged Mmi1 variants, respectively. Mouse anti-DDDDK beads (MBL) were used as controls.

## Fluorescence microscopy

To observe mitotically growing cells, cells were grown in the logarithmic phase in YE or SD liquid medium at the appropriate temperatures. Imaging was performed with the DeltaVision-SoftWoRx system (GE Healthcare). For live-cell imaging, cells were mounted onto glass bottom culture dishes (MatTek, Ashland, Massachusetts) precoated with lectin, and the dishes were filled with liquid medium. For fixed-cell imaging, cells were mounted on coverslips. Images were acquired as serial sections along the $z$-axis at intervals of 0.5 μm (for living cell) or 0.3 μm (for fixed cell). All images were deconvolved and stacked using the quick projection algorithm in the SoftWoRx software.

Single-molecule RNA fluorescence in situ hybridization (smFISH) was performed as previously described (*Castelnuovo et al., 2013*; *Heinrich et al., 2013*) using mixtures of DNA probes (*Supplementary file 3*) coupled to Quaser 570 fluorophore (Stellaris, Biosearch Technologies, Petaluma, California). Probes were targeted against the ORF of the *mei4* gene or the *ssm4* gene.

## Measurement of mRNA spots via image processing

The image processing procedure comprises the following three steps: (1) mRNA spot extraction, (2) nuclei region segmentation, and (3) cell region segmentation.

In step (1), first, local maxima (intensity peaks) in the fluorescence micrographs of mRNAs were detected as the candidates of mRNA spots and the positions of the local maxima were determined. The local maxima were represented as 1-pixel regions via the above process. Second, these local maxima, which have a higher intensity value than the threshold value, were selected as mRNA spots. The threshold value was determined by an empirical analysis based on the distribution of fluorescence intensities in *mei4Δ* micrographs, which were selected as reference (control) samples. For micrograph series of *ssm4* mRNAs, optimum threshold value was determined empirically by visual inspection of each set of micrographs.

In step (2), first, the fluorescence micrograph of nuclei was blurred using a Gaussian filter, and the blurred image was segmented by an automatic local thresholding technique (*Phansalkar et al., 2011*). Second, small isolated regions, which were regarded as noise, were removed by an area opening operation (*Vincent, 1993*).

In step (3), cell region segmentation was performed by a semi-automatic approach. First, the bright field micrograph of cells was blurred using a Gaussian filter to remove noise and smoothen the cell region. Second, edge detection was performed, and the resultant image was binarized. Further, small isolated regions were removed by the area opening operation. Third, a closing operation (*Soille, 2003*) was applied to the binarized image to fill holes and cracks in the binarized cell region. When adjacent or aggregated cell regions were clustered, it was difficult to separate them automatically. Therefore, the contact areas of cells were manually separated, and the shape of the segmented cell region was restored by a region growing technique (*Soille, 2003*) and manual editing.

Finally, we measured the ratio of the number of spots in the nuclei region to that in the cytoplasmic region. This cytoplasmic region is obtained by subtracting the nuclei region from the cell region.

## RNA visualization in living cells

To visualize DSR-containing transcripts in living cells, the human snRNP protein U1A was used, as it specifically recognizes the stem-loop structure of U1 snRNA (*Andoh et al., 2006*; *Takizawa and Vale, 2000*). The constitutive *adh1* promoter, four copies of an U1 snRNA stem-loop sequence, the open-reading frame (ORF) of the firefly luciferase gene (*luc*), and the terminator of the *nmt1* gene were cloned, and tandem repeats of the hexanucleotide DSR motif (TTAAAC) were inserted between the *luc* gene and the *nmt1* terminator. The cloned construct was inserted at the *co2* region on chromosome I. The YFP-tagged U1A protein was expressed from the *arg1* gene locus under the control of the *adh41* promoter, which carried mutations in the TATA sequence of the *adh1* promoter (TATAAATA to ATAAA).

## Yeast two-hybrid

The *mmi1* ORF or its truncated versions were cloned in pGBKT7 or pGADT7 (Clontech, Mountain View, California). The *Saccharomyces cerevisiae* strain AH109 (*MATa, trp1-901, leu2-3, 112, ura3-52, his3-200, gal4Δ, gal80Δ, LYS2::GAL1$_{UAS}$-GAL1$_{TATA}$-HIS3, MEL1, GAL2$_{UAS}$-GAL2$_{TATA}$-ADE2, URA3:: MEL1$_{UAS}$-MEL1$_{TATA}$-lacZ*) was transformed with the plasmids. pGADT7-T-antigen and pGBKT7-p53 were used as controls.

## Electrophoretic mobility shift assay

An RNA electrophoretic mobility shift assay was performed as previously described (*Yamashita et al., 2012*) with modifications. Bacterially purified proteins were preincubated at room temperature with heparin sulfate (5 mg/mL at final concentration) for 20 min to prevent non-specific binding. The proteins were mixed with the RNA probe labeled by digoxigenin (DIG) and incubated at room temperature (23–27°C) for 30 min. Samples were electrophoresed on 6% acrylamide:bisacrylamide (29:1) gel for 3 hr at 150 V and electroblotted to GeneScreen Plus membrane (Perkin Elmer) using 0.5 × tris–borate–EDTA buffer.

## Acknowledgements

We thank S Hauf and D Zenklusen for sharing the FISH protocol; T Tani for providing the RNA visualizing system; D Duncan and J Mata for providing the *mei4-TAP* strain; Center for Radioisotope Facilities, Okazaki Research Facilities, NINS and A Nakade for technical support. This work was supported by JSPS KAKENHI Grant Number 15H04333 to AY, a Grant from The Naito Foundation to AY and a Grant for Basic Science Research Projects from The Sumitomo Foundation (Grand Number 140283) to AY.

## Additional information

### Funding

| Funder | Grant reference number | Author |
| --- | --- | --- |
| Japan Society for the Promotion of Science | 15H04333 | Akira Yamashita |
| Naito Foundation | | Akira Yamashita |
| Sumitomo Foundation | 140283 | Akira Yamashita |

The funders had no role in study design, data collection and interpretation, or the decision to submit the work for publication.

## Author contributions
Yuichi Shichino, Investigation, Writing—original draft; Yoko Otsubo, Yoshitaka Kimori, Masayuki Yamamoto, Investigation, Writing—review and editing; Akira Yamashita, Conceptualization, Investigation, Writing—original draft

## Author ORCIDs
Yuichi Shichino (iD) https://orcid.org/0000-0002-0093-1185
Akira Yamashita (iD) http://orcid.org/0000-0002-1805-1434

## Decision letter and Author response
Decision letter https://doi.org/10.7554/eLife.32155.032
Author response https://doi.org/10.7554/eLife.32155.033

## Additional files
### Supplementary files
• Supplementary file 1. Strains used in this study.
DOI: https://doi.org/10.7554/eLife.32155.026

• Supplementary file 2. Primers used in this study.
DOI: https://doi.org/10.7554/eLife.32155.027

• Supplementary file 3. Oligonucleotide probes for single-molecule FISH.
DOI: https://doi.org/10.7554/eLife.32155.028

• Source data 1. Uncropped images of western and northern blots in *Figure 1C*, *Figure 1—figure supplement 1B*, *Figure 1—figure supplement 2A,B,C,D*, *Figure 3—figure supplement 1A*, *Figure 3—figure supplement 2C*, *Figure 4D,G*, *Figure 4—figure supplement 1D*, *Figure 5C*, *Figure 5—figure supplement 2C*, *Figure 5—figure supplement 3*, *Figure 6C,D,F*, *Figure 6—figure supplement 1C,E*, and *Figure 6—figure supplement 2*.
DOI: https://doi.org/10.7554/eLife.32155.029

• Transparent reporting form
DOI: https://doi.org/10.7554/eLife.32155.030

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
