## [Decision Letter]

Thank you for submitting your article "YTH-RNA-binding protein tethers meiotic transcripts to prevent deleterious translation" for consideration by *eLife*. Your article has been reviewed by three peer reviewers, one of whom is a member of our Board of Reviewing Editors, and the evaluation has been overseen by James Manley as the Senior Editor. The reviewers have opted to remain anonymous.

The reviewers have discussed the reviews with one another and the Reviewing Editor has drafted this decision to help you prepare a revised submission.

The manuscript by Yamashita et al. suggests that Mmi1 tethers meiotic transcripts to nuclear foci and thereby inhibits nuclear export and translation of this class of mRNAs. In addition, the authors illustrate that with the assistance of Erh1, Mmi1 interacts with itself and that this self-interaction is required for nuclear foci formation, target degradation and translation suppression. Overall, the manuscript is well written and the experiments are well conducted. Still improvements are needed as outlined below:

1) A major concern is the way in which the authors talk about the role of Mmi1 in 'translational suppression'. In several parts of the manuscript, the impression is given that this is a direct role of the protein (see for example the title, parts of the Abstract, and in the Discussion). This is misleading. In fact, the authors show that Mmi1 has a role in preventing mRNA export, indicating that any effects on protein levels are likely to be indirect. Indeed, the conclusion on nuclear export in the mmi1 mutant might even be an indirect consequence of defects in Mmi1-dependent nuclear retention and nuclear decay. The question is whether the primary function of Mmi1 is to retain meiotic transcript in the nucleus to prevent cytoplasmic translation and/or to promote nuclear decay at the site of transcription. These issues need to be clarified by more careful interpretation and discussion of the data.

2) Figure 1: The authors use RT-qPCR to measure the levels of *mei4-TAP* and *ssm4-TAP* mRNAs. Yet, Mmi1 was recently shown to influence transcription termination. It would therefore be important to confirm that *mei4* and *ssm4* mRNAs accumulating in mmi1 and red1 mutants have similar 3' ends and poly(A) tail length, both of which are known to influence nuclear export and cytoplasmic translation. Accordingly, the authors should analyze *mei4* and *ssm4* transcripts by Northern blotting and/or RNase cleavage assays to examine the 3' end of these mRNAs.

3) The last part of the manuscript, which describes Mmi1 function in the pairing of homologous chromosomes, is somewhat out of context with the rest of the paper. To tie it better together, the authors could provide more data e.g. by examining the mmi1 domain deletion strains and/or explain/discuss more explicitly why these results are important in the scope of this manuscript. Alternatively, the data could simply be left out of the manuscript.

---

## [Author Response]

1) A major concern is the way in which the authors talk about the role of Mmi1 in 'translational suppression'. In several parts of the manuscript, the impression is given that this is a direct role of the protein (see for example the title, parts of the Abstract, and in the Discussion). This is misleading. In fact, the authors show that Mmi1 has a role in preventing mRNA export, indicating that any effects on protein levels are likely to be indirect. Indeed, the conclusion on nuclear export in the mmi1 mutant might even be an indirect consequence of defects in Mmi1-dependent nuclear retention and nuclear decay. The question is whether the primary function of Mmi1 is to retain meiotic transcript in the nucleus to prevent cytoplasmic translation and/or to promote nuclear decay at the site of transcription. These issues need to be clarified by more careful interpretation and discussion of the data.

We appreciate the reviewer’s comment, and accordingly, have modified the text (including the title, Abstract, Introduction, Results, Discussion and the Figure legends for Figure 1, Figure 1—figure supplement 1, Figure 5, Figure 5—figure supplement 3, Figure 7) to clarify that Mmi1 prevents protein expression of its targets by their nuclear tethering. We consider that the primary function of Mmi1 may be induction of RNA degradation. Accordingly, we have added the following sentence in the revised manuscript: “Induction of RNA degradation might be a primary function of Mmi1”. However, it is difficult to demonstrate this, because we cannot distinguish multiple Mmi1 functions at this point (e.g. by using a mutant that would lose the Mmi1 function, only to tether targets to nuclear foci). We realize that a detailed dissection of the functions of Mmi1 might be beyond the scope of this article. Further study will be required to shed light on this aspect. Nevertheless, the role of Mmi1 in tethering its target transcripts to nuclear foci is apparently important, as shown in the case when Mmi1-mediated RNA degradation is impaired.

We also consider that Mmi1-mediated RNA degradation might primarily occur at Mmi1 nuclear foci, which are distinct from its major target gene loci, because both, target transcripts and the nuclear exosome, localize there. Therefore, we have added the following sentences: “It is possible that Mmi1 co-transcriptionally recognizes its target transcripts at their gene loci and Mmi1-transcript complexes move to the nuclear foci where RNA degradation takes place; however, we cannot exclude the possibility of Mmi1-inducing RNA degradation on its target gene loci. Further investigation would be required to determine the site of RNA degradation and the precise localization of the Mmi1 foci”.

2) Figure 1: The authors use RT-qPCR to measure the levels of mei4-TAP and ssm4-TAP mRNAs. Yet, Mmi1 was recently shown to influence transcription termination. It would therefore be important to confirm that mei4 and ssm4 mRNAs accumulating in mmi1 and red1 mutants have similar 3' ends and poly(A) tail length, both of which are known to influence nuclear export and cytoplasmic translation. Accordingly, the authors should analyze mei4 and ssm4 transcripts by Northern blotting and/or RNase cleavage assays to examine the 3' end of these mRNAs.

According to the reviewer’s suggestion, we have added northern blot analysis (subsection “Mmi1 prevents expression of meiotic proteins in cells with impaired RNA degradation”, Figure 1—figure supplement 2) in the revised manuscript. Transcripts of *mei4* and *ssm4* were similar in size in *mmi1-ts3* and meiotic wild-type cells (Figure 1—figure supplement 2), indicating no significant role of Mmi1 in the termination of both genes. In *red1∆* or *rrp6-32* cells, both *mei4* and *ssm4* transcripts migrated slower than in *mmi1-ts3* cells (Figure 1—figure supplement 2), suggesting that Red1 and Rrp6, but not Mmi1, could have some role in the transcription termination of *mei4* and *ssm4*. These observations also suggest that nuclear tethering of meiotic transcripts, by Mmi1, is independent of the regulation of transcription termination.

In mutant cells with reduced Red1 or Rrp6 function, much longer *ssm4* transcripts were detected (Figure 1—figure supplement 2), which have been shown to initiate from a promoter of an upstream gene and encompass the upstream gene as well as *ssm4* (Zofall et al., Science, 2012).

3) The last part of the manuscript, which describes Mmi1 function in the pairing of homologous chromosomes, is somewhat out of context with the rest of the paper. To tie it better together, the authors could provide more data e.g. by examining the mmi1 domain deletion strains and/or explain/discuss more explicitly why these results are important in the scope of this manuscript. Alternatively, the data could simply be left out of the manuscript.

We agree with the reviewer. As per the reviewer’s suggestion, we have removed the part on homologous chromosome pairing from the revised manuscript. We plan to provide the data related to chromosome pairing in an independent study.